# To Implement A Clear-Water Supply System for Fine-Sediment Experiment in Laboratories

**Qiang Yuan, Man Zhang and Jianjun Zhou \*** 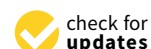

State Key Laboratory of Hydroscience and Engineering, Department of Hydraulic Engineering,
Tsinghua University, Beijing 100084, China; qiangyuan1989@sina.cn (Q.Y.); zhangman86@tsinghua.edu.cn (M.Z.)
\* Correspondence: zhoujj@tsinghua.edu.cn; Tel.: +86-10-62781749

**Abstract:** Fine sediment transport is currently attracting increasing attentions owing to its importance in the dynamics of sediment-contaminant interaction in the fluvial environment downstream of dams, which calls for more detailed and accurate flume experiments. However, because of inaccurate loading and undesired recirculation of fine sediment in the usual short laboratory flumes, such experiments are often unrepeatable and unreliable. In this technical note, we propose a new sediment feeder, to load dry sediment sample at the inlet, and a pressurized sediment filter, to screen the sediment out at the outlet, to implement a clear-water supply system for the flumes. It can improve fine sediment experiments not only by accurate loading but also through preventing undesired sediment recirculation, which can interfere and even modify the designated upstream input conditions. These devices have been constructed and tested, shown to be practical, simple and effective. Using them together can also provide a way to reclaim all the samples of experimental sediment that are of crucial importance for repeat and multiple tests for different contamination with a given sediment without losing the prescribed composition and other properties. This implementation is especially suitable for simulating fine particle affinity contaminant transport in fluvial turbulent flows in low sediment concentrations.

**Keywords:** fine sediment; flume experiment; sediment input; recirculation

## 1. Introduction

Sediment exists widely in natural rivers, which is affected significantly by dams and reservoirs in recent decades. It is regarded as an important problem in hydraulic engineering [1] and also a key component for aquatic ecosystems [2]. Sediment in lowland reaches of large rivers is increasingly altered that very fine suspended sediment is transported in domination as a result of damming. This portion of fine sediment was traditionally regarded as wash-load, which interacts scarcely with rivers and channels [3], so very few attentions had been paid to it in the traditional engineering-oriented sediment studies and understandings on the dynamics of such special sediment in the fluvial environment were inadequate [3–6]. On the other hand, because of the high affinity of many contaminants to fine mineral particles, e.g., phosphorus and nickel, the dynamic circulation, temporal–spatial distribution and final destination of such contaminants are closely related to the fine sediment movement. Especially, the previous wash-load is now becoming the major part of deposition and one of the main medium of contaminants for the relative quiescent water bodies like artificial reservoirs and lakes, as recent human interposition on rivers has changed the paths of sediment significantly [7–9]. Therefore, laboratory experiments on such contaminant with fine sediment in fluvial environment and other aquatic ecosystem attracts more and more attentions [7–13].

Flume experiments as a tool of study have been widely used observing the mechanism of sediment transport in fluvial environment [3,14,15]. As some important particle affinity contaminants are

overwhelmingly adsorbed on and regulated by sediment [7,8,11,12,16], thus flumes should also be used as an effective tool in simulating the mobility and dynamic behaviors of such an eco-matter and contaminant in rivers, along with the transport, deposition and re-suspension of sediment [11,13,17]. However, fine suspended sediment experiments in laboratory flumes is usually inaccurate and hardly repeatable because of the difficulties in accurate loading both in rate and composition and the unavoidable recirculation of the sediment from the outlet to inlet of a flume, which can modify the input conditions significantly in particular for the eco-related fine sediment due to its lightness and long distance adjustment in relative short flumes. Usually, flume sediment experiments are carried out by loading a prescribed rate and composition of sediment at the inlet or through recirculating a given sediment that is pre-laid to the flume. Dry sand load is to deliver the prepared sediment using a feeder connected with vibrating belt [18,19]. Wet loading is to inject prepared sediment slurry to the flume at inlet [20,21]. However, the loading often causes undesired effects on the expected experimental conditions. For example, a long distance bias of vertical distribution of sediment concentration, which is sometimes critical for sediment-contaminant interaction in the flow column. Further, air bubbles adhering to sediment particles is a particular problem for a fine sediment experiment as dry particles are easy to adhere air bubbles that can alter the particles' fall velocity significantly, when they are being submerged into water [22]. Sometimes, it can be observed empirically that many such flocs drift on the water surface throughout a flume if it is not well stirred in turbulent flow. On the other hand, wet loading usually suffers from difficulties in regulation, because the sediment concentration is not easy to be monitored as simple sensors, such as the photoelectric sediment concentration meter, are usually not accurate [13]. Another common drawback of the present loading is that it is difficult to remove the outlet sediment from the recirculation system, especially for fine cohesive sediment during long time test, because laboratory flumes are normally shorter than 20–40 m and settling reservoirs are not big enough to settle out the fine sediment from the system. Owing to such limitations, a substantial portion of the sediment can run through the reservoir and be re-input to the flume inlet again, that can deviate the inlet sediment condition. For instance, the recirculating sediment for a normal flume experiment was measured in excess of 80%–90% of the total load when sediment is finer than 0.05 mm with the background concentration being elevated constantly with time [13]. On the other hand, repeating tests using the same sediment may yield amply different results as some portion of the sediment sample may stray away during the recirculation, especially when large quantities of graded fine sediment sample are encountered. Both defects in accuracy and repeatability in present flume system are inevitable due to the objective limitations in setups. This is a lasting problem in laboratory experiment for fine suspended sediment [23], which appears also as one of the major reasons for the unreliability of fine sediment studies that are based on flume experiments [3,24]. Better and reliable simulations for the sediment–contaminant interaction in flowing water needs innovative implementations.

In this technical note, we propose a new sand feeder and an innovative sediment filter to implement a clear-water supply system for fine sediment flume experiment in laboratories. It is aimed to improve the loading accuracy and reliability for with particular emphases to fine sediment experiment, to provide a readily fundamental experimental method for simulating fluvial fine sediment transport and its interaction with the contaminant and other eco-related matters in laboratories.

## 2. System Description

The proposed clear-water supply system, based on conventional laboratory flumes, is implemented by adding a dry-sand feeder at the inlet and a pressurized sediment filter at the outlet of the flumes, and a practical example is shown in Figure 1.

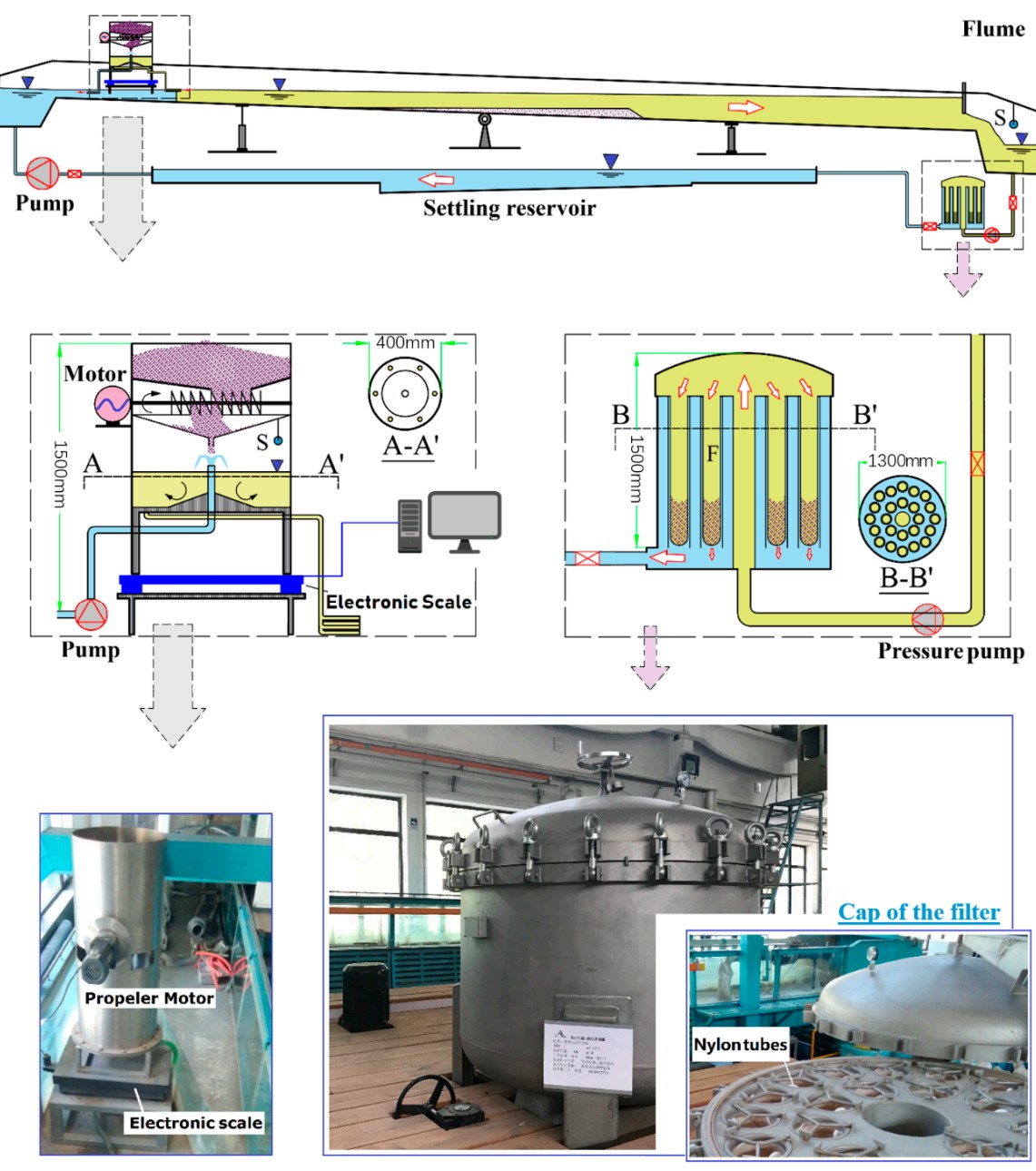

**Figure 1.** Illustration of the proposed clear-water supply system with the feeder (**left**) and filter (**right**) constructed and used in the Laboratory of Sediment Research, Tsinghua University. In the figure, the yellow and blue colors represent muddy and clear water respectively, and the symbol S refers to water level sensors; the maximum rotational speed of adjustable motor for propeller is 1300 rpm; the maximum discharge of the clear water-pump for the feeder is 2 L/s.

The feeder is an integrated device that is composed of a dry-sand hopper at the top, a motor driven propeller, a clear-water pump, a water-sediment mixing chamber with a rotating flow to deliver well-mixed water-sediment flow to the flume through six plastic pipes, and an electronic scale at the bottom. The pump generates an upward water jet in the chamber. Dry-sediment without weighing can be input to the hopper, which is driven to the lower chamber by the propeller and sediment runs into the water jet and it is mixed with water in the chamber. Finally, a sediment-laden flow is injected into the flume through six symmetrically distributed flexible pipes. Herein, the bottom of the chamber is designed as an up-convex cone that can avoid sand accumulation. The feeding rate of dry sediment

from the hopper is controlled by the propeller's rotating frequency, which is calibrated beforehand and is thus adjustable according to the rotating speed of the motor. The sand feeder sits on an electronic scale, which is connected to a computer for data recording. Before an experiment, the water level in the chamber must be stabilized to an appropriate height. Then, a well mixed sample of dry sand can be added to the hopper without being weighed. The weight of the sediment in the hopper together with water in the chamber are both recorded automatically. The net rate of sediment input can be calculated by the computer according to the weight variation. Since the pump takes sand-free water from the upstream side of the flume, the loading does not affect the prescribed flow discharge and sediment rates in the experimental section. To reduce the influence of water-level fluctuation, it is better to add a sensor to monitor the water level in the chamber so that the error from water weight variation can be excluded. A noticeable advantage in using this feeder is its ability in elimination particle wearing bubbles as fine sediment has been stirred sufficiently in the turbulent flow chamber. This is important especially for fine sediment experiments.

The pressurized filter is fixed inside of a sealed steel tank (Figure 1). It is a separate equipment that can be attached to any flume if it is needed. The filter consists of 24 membrane tubes to screen and to collect sediment. The tubes are attached to 24 orifices. The membrane is made of nylon, and the tubes are originally made as filtration membrane for waste water treatment. It is widely available. The pore size of the membrane is selected according to the minimum size the sediment to be filtered. The 10 μm membrane we used in this case can screen all of the sediment coarser than 8 μm. The tank is connected to the flume outlet via a pipe attached to an ordinary water pump to collect the water and sediment at the outlet and to generate the pressure for the filtration. After the tube filtration, water with some very fine sediment, e.g., finer than 8 μm, is refilled back to the settling reservoir through a pipe. The size of filter, including the pump and number of tubes, should be selected based on the scale of flow discharge and rate of sediment the experiment encountered. For some experiments, more than one filter or one filter with many finer membrane tubes can be considered when a large amount or very fine sediment is encountered.

As all the concerned fractions of sediment have been screened out and eliminated from the circulation system, only clear water returns to the inlet. Therefore, loading accuracy of sediment at the inlet depends solely on the sand feeder. In this way, all of the prepared samples of sediment both deposited in the flume and trapped in the tubes can be collected. Thus, for the first time, it is possible to reuse the same sample of sediment to run a series of parallel experiments. It is especially meaningful for contaminant transport in different conditions with the same composition and other basic properties of fine sediment. This is of great importance for graded fine sediment as accurate running of parallel tests using identical sediment samples are almost impossible previously. In addition to the improvement in inlet loading and recirculation for sediment concerned experiments, accuracy and reliability of sediment associated contaminant flume experiments in laboratories can also be enhanced.

## 3. Tests and Applications

The proposed dry-sand feeder and sediment filter were constructed and tested in the Laboratory of Sediment Research, Tsinghua University. They were connected to a long flume of 64 m in length and 0.4 m in width and a wide flume of 40 m in length and 3.5 m in width. The test equipment and their installations are illustrated in Figure 1. For this case, the feeder was made from a section of 0.4-m-diameter and 1.5-m high stainless pipe. In the narrower flume experiment, flow discharge was 20–30 L/s and sediment concentration could be set to 1–2 kg/m$^3$. The selection of flow and sediment discharges was based on the nature flow velocity and turbulence conditions the test aimed at.

An example of the feeding test and system application can be illustrated through an experiment of fine suspended sediment deposition in a fluctuating backwater reach of a reservoir. The experiment was to simulate very fine sediment that was coarser than 8 μm. It is almost impossible to run such a fine sediment experiment in flumes previously, as the fine portions of sediment were either recirculated in a big ratio with the water flow, which would be either added to the inlet modifying the inlet

sediment condition, or trapped considerably into the underground settling reservoir, and making the fine sediment less and less from run to run, before using this system. In addition, some inlet-loaded dry sediment tended to float in a long distance as flocs in the flow due to the effect of air bubbles, thus incorrectly distorted the formation of deposition in the experiment. Figure 2 illustrates the rate of sediment load input calculated using the data recorded for a 4-h process. It can be seen that the input sediment load is stable, except in the initial few minutes when some sediment accumulated in the chamber. Although the data showing some fluctuations, the standard deviation of sediment load is 2.1%, which is 0.675 g/s in sediment load. The fluctuation is due to a slight variation in the water level inside the chamber of the feeder, which can be corrected if a water-level sensor is used inside the chamber (Figure 1). The rate differences between the feeder input (recorded) and that according to the actual input sediment were generally less than 0.5% for the whole sediment feeding process.

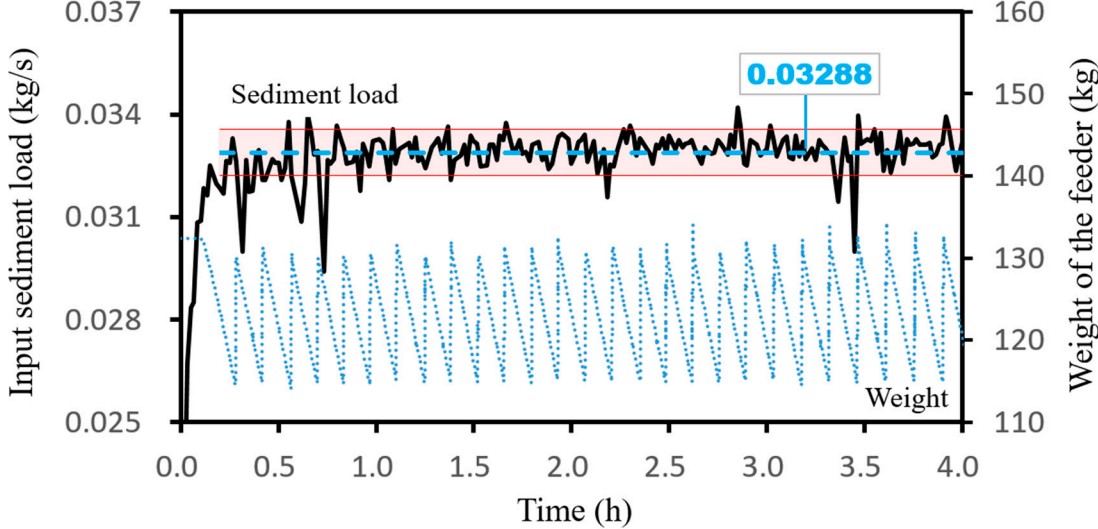

**Figure 2.** Sample record of the data from a sediment experiment when the flume is equipped with the dry-sand feeder. Recording frequency of weight is 1s. The rate of sediment load (black line) is calculated according to the feeder's recorded weight (dotted blue line) using 1-min averages, it agreed with the real value (blue number). The pink band indicates the limit of standard deviation.

To check the efficiency of the filter, data from an experiment using very fine kaolin sediment, with a median laser diameter (laser diameter in this note denote the sediment diameter measured by laser particle-size analyzers, for instance, LA960 that is generally larger in number than the traditional setting diameter, which is generally used in sizing the fall velocity of sediment particles based on formulas of Stokes and others [3]. These two diameters can be converted using a relationship we obtained in the present note, i.e., $D_{setting} = 0.451 D_{laser}^{1.07}$ (diameter units in μm), as indicated in Figure 3) $D_{50} \approx 10$ μm (approximately 5-μm in setting diameter), were introduced here. In this experiment, the flow discharge used in the 60 m long and 0.4 m wide flume was 20 L/s and the deposition of sediment with a setting diameter fine to 8 μm (15 μm laser diameter) was required. Since a considerable amount of cohesive sediment could run off the system, the recirculated fine sediment not only changed the rate but also even deviated the fall velocity of the sediment input due to the effect of flocculation. For this purpose, the pore size of membrane for the filter was chosen as 10 μm and sample of experiment sediment were loaded using the dry-sand feeder. The loaded sediment, the flume-deposited sediment (that collected and mixed up together) and the filter-trapped sediment were all sampled and analyzed separately to determine their fractional distributions. Figure 4a depicts the measured percentages of sediment for seven size fractions from 1 to 300 μm. It indicates that coarser fractions were mostly deposited in the flume and trapped by the filter, some fine portions of setting diameter less than 10 μm were trapped by the filter while a large ratio of them escaped. Nevertheless, all the sediment with a

setting diameter coarser than 8 μm were all screened out and reclaimed by the filter. After thorough mixing of the claimed sediment, distribution of sediment in the required range (laser diameter coarser than 15 μm) was almost identical to that of the inlet input (Figure 4b). This result indicates a high efficiency of the filter that no sediment in the required range could escape and recirculate with flow into the inlet through the system. A clear inlet flow was thus produced and the prepared sample of sediment could well be conserved, to enable reusing the samples with the prepared composition multiply for comparable experiments.

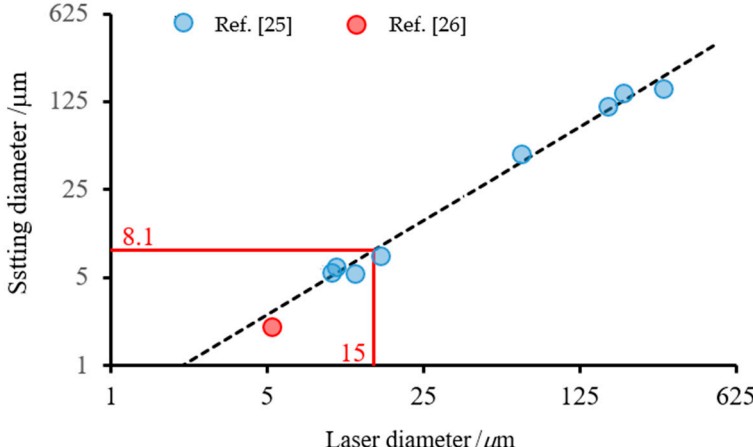

**Figure 3.** Relationship between laser and setting diameters using experimental data of [25,26].

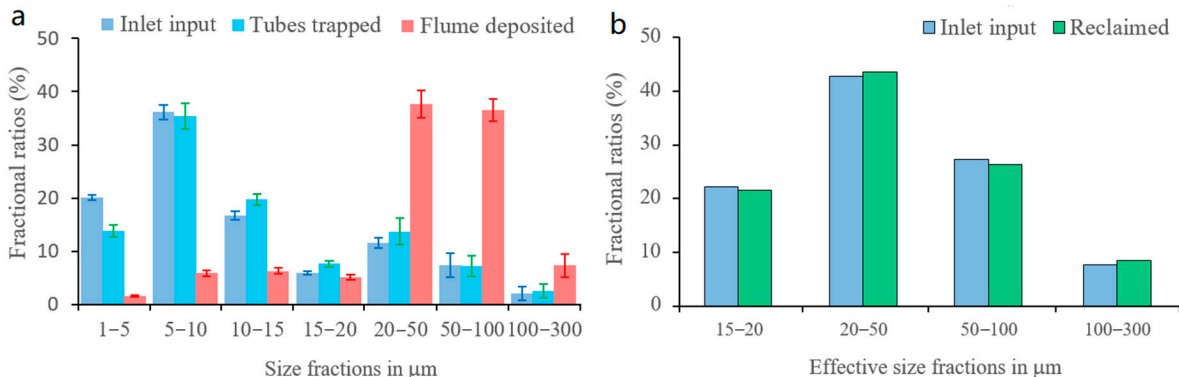

**Figure 4.** Measured fractional distributions of the sediment from a flume experiment of a very fine kaolin transport in the Laboratory of Sediment Research, Tsinghua University. (**a**) Full-size range distribution of the input, filter-trapped and flume-deposited sediment with their variation at a 95% confidence interval. (**b**) Comparison of the average sediment composition in the effective size range (coarser than 15 μm laser diameter) between the inlet input and the reclaimed sediment that are mixed up in the flume and filter tubes.

The aforementioned preliminary tests and applications suggest the proposed feeder and filter can be used as key facilities for a clear water supply system, which is instrumental in improving the accuracy of present fine suspended-sediment flume experiment in laboratories.

## 4. Conclusions

In this technical note, a new feeder and a pressurized filter were proposed and constructed for flume sediment experiment in laboratories. The main aim of the implementation was to improve the long and well applied laboratory flume system for a suspended sediment experiment with better loading and recirculation. It is especially proposed for the very fine sediment that was regarded as "wash-load" traditionally, therefore, to provide an effective method for simulations of a sediment–contaminant

interaction in nature rivers. Detailed suspended sediment deposition experiments were carried out and the efficiencies of the proposed facilities were tested. The mean findings were:

(1)　Sediment loading accuracy by the dry sand feeder could be obviously improved with standard deviation and absolute error in loading rate smaller than 2.1% and 0.5%, respectively.

(2)　The filter supported water supply system could provide almost clear inlet water flow, free of sediment in the required sediment range, and conserve the experiment sediment samples with almost identical composition between the input and the reclaimed sediment, which could be used readily for parallel experiments with less uncertainties from sediment. It is a reliable way to be used for fine sediment encountered fluvial environmental studies with enhanced repeatability by means of experimental sediment sample conservation.

(3)　These devices working together could provide much better condition for fine sediment and for sediment–contaminant interactions in laboratory flumes, owing to the merits in accurate loading as well as an effective screening of sediment in the recirculating system.

　　Therefore, laboratory flumes could now be more feasibly used in simulating a fluvial sediment–contaminant interaction and their related eco-environmental processes under dynamical flow conditions in addition to field observation, where fine sediment dominates.

**Author Contributions:** Q.Y. contributed to the flume experimental tests, analysis and draft manuscript preparation; M.Z. contributed to the design and installations of the dry sand feeder, the pressurized filter and the system implementation; J.Z. contributed to the conceptual idea, technical layout, supervision, project administration, funding acquisition, formal analysis and formal manuscript writing.

**Funding:** This research has received funding from the National Key Research and Development Project of China under grant agreement No 2016YFE0133700, supports from the State Key Laboratory of Hydro-science and Engineering, Tsinghua University under grant No 2018-ky-06 and the Comprehensive Surveys of Natural Resources and Environment in Typical Spatial-planning Areas of China under grant No DD20190506.

**Acknowledgments:** The authors are grateful to B. Lin for his kind support for key experiment facilities.

**Conflicts of Interest:** The authors declare no conflict of interest. The funders had no role in the specific investigation and in the decision to publish the results.

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
