# Peer review of "To Implement A Clear-Water Supply System for Fine-Sediment Experiment in Laboratories"

_water, doi:10.3390/w11122476_

Round 1
Reviewer 1 Report
I believe the manuscript can be improved. The authors need to compare results with similar studies which have been conducted in the past. Some of the strong statements which were made in the introduction section should have references (e.g., L55-L 65, L67-L73).
It will be useful to see how results compare between modified flume design and original flume design.
What was the rationale behind selecting the flow discharge value?
Overall I think manuscript can be substantially improved.
Author Response
1) we have substantially revised the the introduction and conclusion;
2) In the revised introduction, we have added references as more as we can, and we also added some more explanations to clarify the so called "strong statement". But, it is also a fact that the under discussed defects on the traditional flume experiment have been for a long time well known and well ignored in the field of sediment research. Therefore it is difficult for us to find more references that had discussed the problems compactly.
3) We added a currently published result (Huang et al 2015) of experiments that carried almost unacceptable recirculation of sediment from their flume experiment for fine sediment of diameter less than 0.05mm, that implies the present proposed system can improve such experiment fundamentally as we can provide clear water to the inlet.
4) We added one more sentence "The selection of flow and sediment discharges is based on the nature flow velocity and turbulence conditions the test aimed at" to indicate the rationales for the flow discharge selection.
5) Although we have tried our best in improving the manuscript's English, but it may be still not enough. So we would like to ask, with appreciates, the help from MDPI English editing service if it is still needed. We will pay fee for the editing.
Reviewer 2 Report
The work is very interesting and well good. I have no negative comments.
Author Response
Thanks
Round 2
Reviewer 1 Report
The authors have addressed my previous comments satisfactorily. However I feel some sentences are still not structured properly.
This manuscript is a resubmission of an earlier submission. The following is a list of the peer review reports and author responses from that submission.
Round 1
Reviewer 1 Report
The work is very interesting and well good. I have no negative comments.
Reviewer 2 Report
In this manuscript, the authors describe the pollution/contamination issues arising from the discharge of very fine suspended sediments through dammed rivers and the inadequacy of current experimental laboratory techniques. They have constructed and tested a flume for the study of fine-sediment-chemical interactions that adequately addresses those issues.
I am afraid that, while the scope of the manuscript renders it appropriate for publication as a technical report in Water, I can’t recommend its publication because of seriously flawed language. It includes many expressions which, even though not grammatically wrong, are unusual and awkward. I select here a few expressions from the abstract: “its dominations in lowland rivers”, “are revealed to be accurate”, “guarantees better input”, “for reuses”, etc. In addition, issues with tenses, singular/plural use, punctuation, etc. abound in the manuscript. A spell-checker is unlikely to capture the dozens of expressions that need to be changed. Therefore, I suggest that the authors screen their manuscript again and select better terms, have a native-English-speaking colleague help them out, or seek the help of a professional editor.
Secondly, suspended-sediment flumes are actually not that unusual. I do think that the facility designed by the authors is worth reporting on, but an explicit connection must be made between the specifications of the designed facility and the rates of chemical interactions with suspended sediments. I assume flow must be pretty slow but realistic, and therefore the design can perform that but it must be elaborated more explicitly. For instance, what is the residence time of water and suspended sediment in the flume during the kaolinite experiment (in line 119, we are told that the pump used at the test facility has a maximum discharge of 2 L/s, so that would set the lower residence time)? Was the kaolinite experiment conducted at that maximum discharge or slower, and what are some recommended specifications for the fine-sediment-chemical interactions studies the facility is designed for?
I also include some points of lesser importance below for the authors’ consideration.
It would be more sensible and practical to combine figures 1 and 2 in a multi-panel figure, that can occupy a page, if necessary. Please address the following:
In Figure 1, numbers are distances, I presume. What are the units? Indicate many of the components you mention in the text (e.g., in ines 77-78) on the diagrams themselves.Be more specific about instrumentation and parts. I do understand that many are custom-made and others are common in the industry. Please state which are which, and be as technical as possible with your definitions. For example:
Line 97: “Nylon-membrane tube with appropriate pore size”. State the pore size here. Line 98: “Ordinary pump”. What is meant by “ordinary”, i.e., what is the pump type?The phrase “entire sediment” (ines 105, 107) doesn’t make sense. How can it be separate from the circulation system AND be deposited in the flume? Please explain or use a different term.
I assume that the text in lines 129-135 is not a caption but a regular paragraph. It should be formatted as such.
In Figure 3, add the standard deviation you mention in lines 131-132 in the figure, using a +/- symbol after the average value of the sediment load you indicate.